# The effects of school-based hygiene intervention programme: Systematic review and meta-analysis

**Sophia Rasheeqa Ismail**[1]*, **Ranina Radzi**[1], **Puteri Sofia Nadira Megat Kamaruddin**[1], **Ezarul Faradianna Lokman**[1], **Han Yin Lim**[1], **Nusaibah Abdul Rahim**[2], **Hui Yin Yow**[2], **Daarshini Arumugam**[3], **Alex Ngu**[3], **Annie Ching Yi Low**[3], **Eng Hwa Wong**[4], **Sapna Patil**[4], **Priya Madhavan**[4], **Ruslin Bin Nordin**[5], **Esther van der Werf**[6], **Nai Ming Lai**[4]

1 Nutrition, Metabolic and Cardiovascular Research Centre, Institute for Medical Research, National Institutes of Health, Ministry of Health Malaysia, Shah Alam, Selangor, Malaysia, 2 Faculty of Pharmacy, University of Malaya, Kuala Lumpur, Malaysia, 3 Ministry of Health, Malaysia, 4 School of Medicine, Faculty of Health and Medical Sciences, Taylor's University, Subang Jaya, Selangor, Malaysia, 5 Faculty of Medicine, Bioscience and Nursing, MAHSA University, Jenjarom, Selangor, Malaysia, 6 Homeopathy Research Institute, London, United Kingdom

* sophia.rasheeqa@moh.gov.my

**Data Availability Statement:** All relevant data are within the manuscript and its Supporting Information files.

## Abstract

Children are susceptible to infections due to frequent participation in school group activities and their often-suboptimal hygiene practices. Frequent infections in children affect school attendance, academic performances, and general health. The effectiveness of school-based hygiene-related intervention programmes need to be informed by updated high-quality synthesised evidence. In this systematic review, we searched PubMed and Cochrane CENTRAL for randomised and non-randomised interventional studies that evaluated school-based hygiene-related interventions. We assessed risk-of-bias (Cochrane risk-of-bias 2 tool), performed random-effect meta-analysis (RevMan 5.4) and rated certainty-of-evidence (GRADE). Thirty-nine trials (41 reports), published from 2011 to 2024 from 22 countries were included. Twenty-three studies contributed data for meta-analysis. All school-based interventions were compared with standard curriculum and demonstrated very low to low certainty-of-evidence due to study methodological limitations and imprecision. Hand-body hygiene interventions may improve knowledge, attitudes and practices (SMD 2.30, 95%CI 1.17 to 3.44, 6 studies, 7301 participants), increase handwashing practices (RR 1.75, 95%CI 1.41 to 2.17, 5 studies, 5479 participants), and reduce infection-related absenteeism (RR 0.74, 95%CI 0.66 to 0.83, 5 studies, 1017852 observations). Genital hygiene interventions may improve attitude (SMD 6.53, 95%CI 2.40 to 10.66, 2 studies, 2644 participants) and practices (RR 2.44, 95%CI 1.28 to 4.68, 1 study, 1201 participants). However, intervention effects on oral hygiene appeared mixed, with worsening of the oral hygiene score (SMD 3.12, 95%CI 1.87 to 4.37, 2 studies, 652 participants) but improved dental hygiene (SMD -0.33, 95%CI -0.53 to -0.13, 3 studies, 4824 participants) and dental caries scores (SMD -0.34, 95%CI -0.52 to -0.16, 4 studies, 2352 participants). Limited evidence suggests that interventions targeting hand-body and genital hygiene practices may improve knowledge, practices, and infection-related absenteeism. However, the effects on

**Funding:** The author(s) received no specific funding for this work.

**Competing interests:** The authors have declared that no competing interests exist.

oral hygiene intervention appeared mixed. Future research should strengthen randomisation and intervention documentation, and evaluate hygiene-related behaviour, academic performances and health outcomes.

## Introduction

Personal hygiene is essential for maintaining optimal health and preventing the spread of diseases. It encompasses a range of practices, including daily body cleansing, regular tooth brushing, handwashing with soap, nail grooming, wearing clean attire, and adhering to proper cough and sneeze etiquette [1]. Young children often exhibit suboptimal hygiene habits, making them more susceptible to communicable diseases [2–5]. Global prevalence of respiratory infections and diarrhoea remains a significant concern, with the United Nations Children's Fund (UNICEF) identifying them as leading causes of mortality in children under the age of five, accounting for over 2 million deaths worldwide annually [6]. Infectious diseases in childhood often lead to increased school absenteeism, poor academic performance and health status [7–10]. Specifically, childhood diarrhoea has been linked to long-term cognitive deficits [11].

The Coronavirus Disease 2019 (COVID-19) pandemic spotlighted schools as potential disease transmission hubs, partly due to students' inadequate knowledge and personal hygiene practices [12]. School-to-home transmission of COVID-19 increased containment challenges, leading to suboptimal disease management [13] and an upsurge in reported antibiotic resistance, particularly in low- and middle-income countries [14–16]. However, the pandemic has also reignited interests in the role of hand hygiene in disease prevention. For example, Cambodia reported a 20% increase in the number of group handwashing facilities, with secondary schools witnessing the most significant improvements [17]. On the other hand, nationwide surveys in Ecuador and Sudan in 2020 and 2021 respectively raised concerns that only half of the schools had basic hygiene services or facilities, with only a minority in satisfactory condition [18].

School-aged children dedicate a substantial portion of their time to group activities in school, and this poses a risk of communicable diseases transmission. Multifaceted approaches in national-level hand hygiene campaigns across Europe have been observed, however, these campaigns had a particular emphasis on healthcare professionals [19].

There is a relative lack of emphasis on education concerning daily hygiene practices at home and at school. Global and national infection control and antimicrobial resistance (AMR) strategies have traditionally focussed primarily on healthcare facilities rather than schools [20, 21]. It is imperative to instil comprehensive awareness and knowledge of hygiene practices, including judicious antibiotic use, among students through dedicated health education programs integrated into the school curriculum, with active involvement of educators and designated healthcare personnel. Such preventive strategy is likely more effective than the use of antibiotics in treating infections when they occur. Educational resources, such as the e-bug campaign operated by the UK Health Security Agency, provide free materials for schools to educate students on hygiene, infections, and antibiotics [22].

In addition to health education, the availability of essential hygiene facilities, such as handwashing stations with clean water, is vital for promoting and sustaining good hygiene practices. The WHO/UNICEF Joint Monitoring Programme (JMP) underscores the necessity to extend water, sanitation, and hygiene (WASH) facilities beyond households, with the initiation of WASH in Schools (WinS) project, which has been piloted in countries such as Belize,

Guatemala, Honduras, India, Indonesia, the Philippines and Kenya [23, 24]. The projects focus on building capacity in training teachers to incorporate hygiene programming into classroom lessons and school amenities. Examples included using the school's sanitation facilities as a site for interactive and integrated practical learning, for example, supervised handwashing sessions with soap before eating and after using the toilet [18]. The intended result is that children acquire knowledge about disease transmission and skills in good hygiene practice, which they can take home to their families [23, 24]. Studies have shown that the availability of handwashing facilities is associated with reduced reported illness and absenteeism and increased hygiene knowledge among school children [18, 25, 26].

Mass education in promoting personal hygiene awareness and practices is widely considered essential in safeguarding the health of the younger population, although the extent of its effectiveness in improving knowledge and practice among different settings, evaluated through rigorous studies, remain unknown. We aimed to synthesise the effectiveness of school-based hygiene-related intervention programmes, specifically to look into its impact on improving knowledge, practice and reducing infection-related morbidities.

## Methodology

The systematic review is registered with The International Prospective Register of Systematic Reviews (PROSPERO) (CRD42021253638).

### Eligibility criteria

We included randomised and non-randomised controlled trials, cluster randomised controlled trials, and crossover trials. Studies conducted in preschool, primary school, secondary school, and general college students were included. Studies conducted in healthcare-related colleges or among healthcare students (such as student nurses and dental nurses) were excluded as changes in outcomes do not represent that of the general population.

We included any form of intervention (educational, behavioural, and structural) implemented to improve hygiene-related outcomes, compared with no intervention or any other form of intervention that aimed to improve hygiene-related outcomes, as detailed below.

### Outcome measures

1. Level of knowledge on personal hygiene including characteristics and function of hand hygiene, body hygiene, oral hygiene, and menstrual/genital hygiene. We accepted any tools used by the authors of the individual studies that covered awareness, knowledge, and practices of hygiene in combinations or separately.

2. School absenteeism due to infections or sickness, in particular communicable diseases. We accepted medically documented and self-reported causes of school absenteeism as well as repeated reports for the same student. The types of infections included respiratory infections (characterised by symptoms such as sore throats, coughs, asthma, and influenza-like illness), gastrointestinal infections (characterised by symptoms such as diarrhoea and vomiting), and unspecified infections requiring prescription of antibiotics or other treatments.

3. Changes in oral hygiene status using standard measurement tools such as the Oral Health Index Simplified (OHI-S) score, plaque score, modified gingival index (MGI), and others.

4. Changes in dental status using standard measurement tools such as Decayed-, Missing-, Filled Teeth (DMFT), Decayed-, Missing-, Filled Surfaces (DFMS), and others.

5. Changes in dental caries using standard measurement tools such as International Caries Detection and Assessment system (ICDAS II) and others.

## Search strategies

We searched for relevant studies in MEDLINE (PubMed), Cochrane Central Register of Controlled Trials (CENTRAL) (which covers EMBASE and CINAHL for published studies, and trial registries including the WHO International Trial Registry Platform and ClinicalTrials.gov for ongoing studies). Latest database search was performed on 16th April 2024. No language and publication period restrictions were applied. Relevant clinical trials and conference abstracts were reviewed and searched. The search strategy is summarised in S1 Appendix.

## Selection of studies, data extraction and management

Titles and/or abstracts were independently screened in duplicates by three pairs of authors (SRI & RAR, LHY & AN, ACYL & DA) for potentially eligible studies. Four authors (SRI, RAR, LHY & EFL) subsequently evaluated the shortlisted abstracts and full texts to determine eligibility. We outlined our study selection process in a PRISMA diagram (Fig 1). Three authors (RAR, LHY & EFL) independently extracted all data from each included study using a standardised data collection form which included study design, study population, intervention details, comparison details, and outcomes. All extracted data were evaluated by SRI, EFL and PSNMK for accuracy.

## Risk of bias assessment

The Revised Cochrane risk-of-bias tool for randomised trials (RoB 2) was used to assess the risk of bias in individually randomised, parallel-group trials with regards to the results of the main outcomes. There are five domains explored in this RoB 2 tool: risk of bias arising from the randomisation process (Domain 1), risk of bias due to deviations from the intended interventions (Domain 2), risk of bias due to missing outcome data (Domain 3), risk of bias in the measurement of the outcome (Domain 4), and risk of bias in the selection of the reported result (Domain 5). Within each domain, there are a series of signalling questions with multiple response options (Yes/ Probably Yes/ Probably No/ No/ No Information) to elicit information that is relevant to the assessed domain. Additional considerations arising from period and carryover effects were recommended for crossover trials [27]. The Revised Cochrane risk-of-bias tool for cluster-randomised trials (RoB 2 CRT) was used to assess the risk of bias in cluster-randomized trials. This tool contains six domains. In addition to the five domains in RoB 2, Domain 1b assesses the risk of bias arising from the timing of identification or recruitment of participants [27]. Two review authors (SRI and RAR) assessed each included trial. Each domain was judged as 'Low', 'Some concerns', or 'High' risk of bias based on the answers to the signalling questions algorithm. In general, studies were judged as 'Low Risk of Bias' if they met the criteria for low risk in all domains, 'Some Concerns' if there were concerns in at least one domain, and 'High Risk of Bias' if there was high risk in at least one domain or multiple domains with some concerns. The overall summary of the risk of bias assessment of included studies is reported in S3 Appendix.

## Data synthesis

Outcomes were extracted and included in the quantitative analysis for the same comparisons. For the knowledge, attitudes, and practices of hand hygiene outcomes, all measurement tools

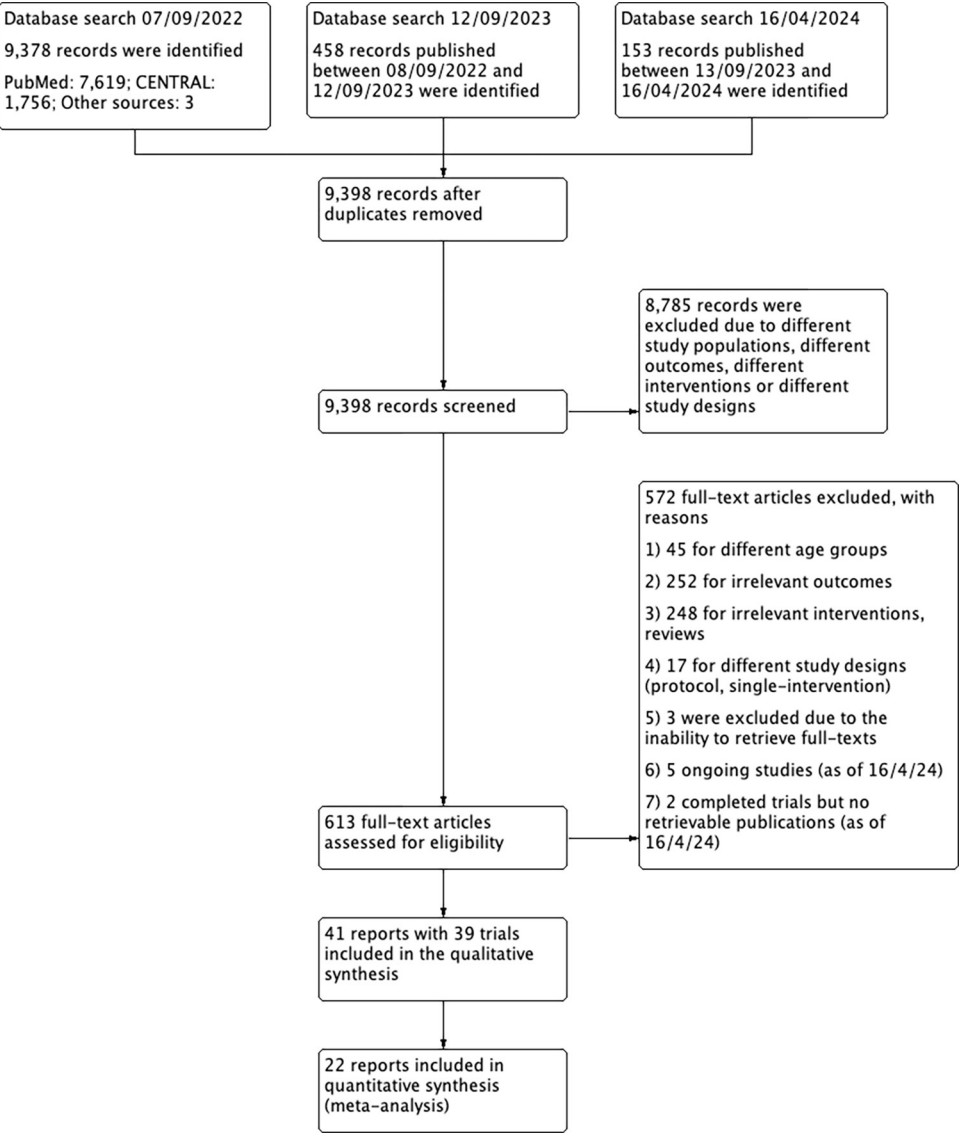

**Fig 1. PRISMA flowchart of included studies.**

with the same outcome were included. For studies that did not report standard deviations (SD), the SD were calculated from the reported confidence intervals (CI) or from given quantile values. The denominator in the measurements for handwashing with and without soap practices was the total number of observed target events. The denominator used in the school absenteeism measurements was the total number of possible absent days/weeks.

For outcomes measured by questionnaires or scores, mean differences (MD) of the total points after intervention were used as the summary measure. For event-based outcomes, risk ratios (RR) were used as the summary measure. MD, RR, and corresponding 95% CI were calculated based on the extracted values and denominators described earlier, with the RR being the ratio of the intervention over control or MD being the difference in total score between the intervention and control. Results were pooled using random effects meta-analysis using the Mantel-Haenszel method for analysis to accommodate the differences across the studies (RevMan 5.4). Absolute risk values were calculated with GRADEpro GDT software. Risk of bias,

inconsistency, indirectness, imprecision, and publication bias were used for the GRADE considerations. We justified all decisions to downgrade the certainty of evidence in the footnotes. To further explore the differences across different comparisons, we performed several subgroup analyses. In hand hygiene interventions, we explored the relative effect of studies which had interventions lasting more than one month. In oral hygiene intervention comparisons, we explored the relative effect of studies conducted in different settings.

## Results

### Study selection

We included 41 reports from 39 intervention studies (Fig 1). Larsen 2020 and Ryom 2022 were from the same intervention trial [28, 29]. Wu 2017 and Wu 2021 were also from the same trial [30, 31]. Among the included trials, 25 evaluated hand hygiene interventions, four evaluated genital hygiene interventions, and 12 evaluated oral hygiene interventions (S1–S3 Tables respectively) (Fig 2). Only one study evaluated outcomes on antimicrobial resistance [32].

### Study characteristics

Four out of 39 trials were conducted in a single school [33–36], while others included multiple schools. The smallest study was conducted in 60 children with autism in Belagavi, Karnataka, India to evaluate oral hygiene practices using a Picture Assisted Illustration Reinforcement (PAIR) communication system as compared to standard curriculum [37]. Whilst the largest included study was conducted in 60 primary schools (n = 44,451 students) in Cairo, Egypt

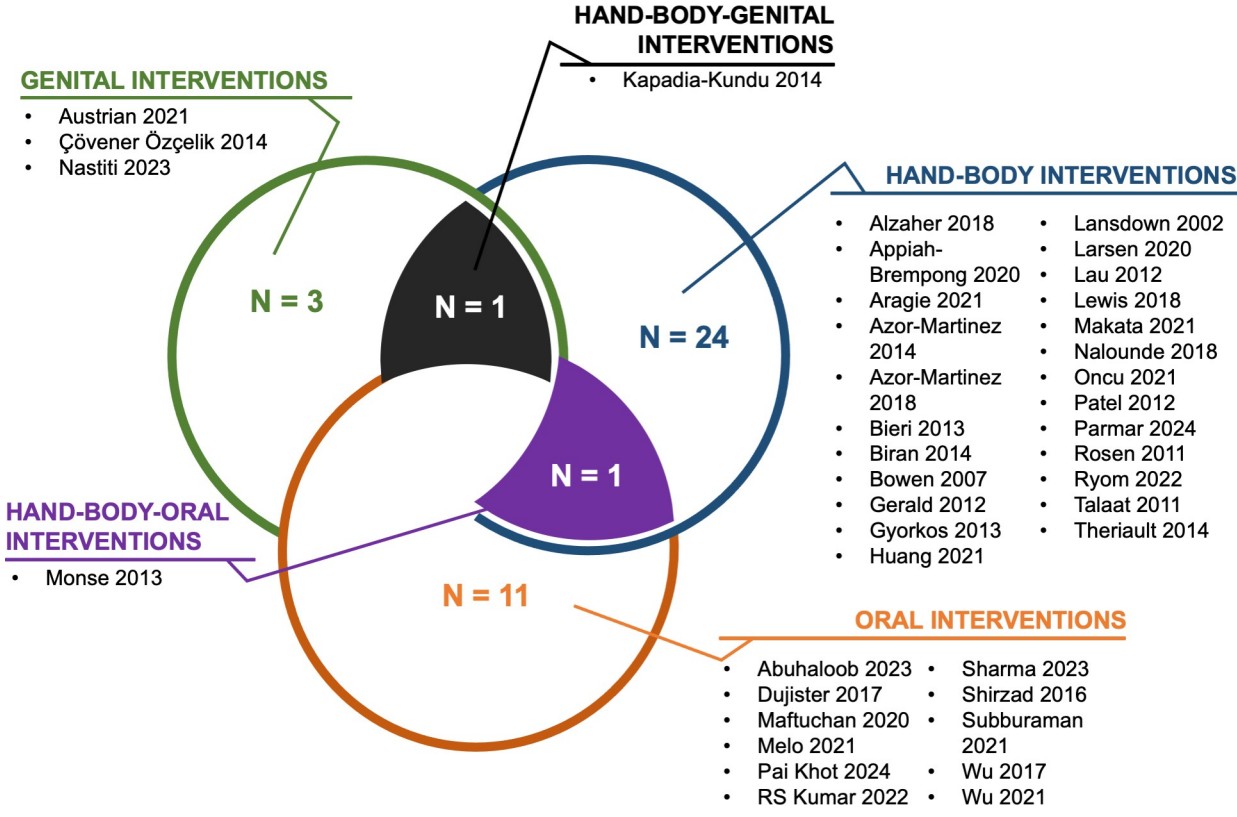

**Fig 2. Summary of included studies by intervention.** N is the number of represented studies.

[38]. Five studies were conducted in pre-schools, 25 in primary schools, and 12 in secondary schools or colleges.

## Intervention components

All trials (n = 39) evaluated intervention programs that included behavioural and/or educational components, and eight studies evaluated concurrently the effects of structural changes to the school facilities [8, 39–45]. Behavioural interventions included tasks that were performed repeatedly, exercises, games, puzzles, and other forms of reinforcements. Educational interventions included information-sharing sessions, talks, and lectures. Structural interventions included the construction of washing facilities such as a handwashing water station near latrines for handwashing and near classrooms for drinking and the installation of liquid soap and hand sanitiser dispensers. We summarised the interventions conducted by each study in S1 Table for hand hygiene interventions, in S2 Table for genital hygiene interventions, and in S3 Table for oral hygiene interventions.

## Geographical representation of included studies

Fig 3 summarises the locations of the included studies. The studies were conducted in 22 countries, including low- and middle-income countries such as India (n = 8), Indonesia (n = 4), China (n = 4), Philippines (n = 2) and Kenya (n = 2), and in developed countries such as the United States of America (n = 2), Spain (n = 2), and Denmark (n = 2).

## Risk-of-bias assessment

Among 34 cluster-RCTs, 10 were judged to have an overall high risk-of-bias, and 24 were judged as having some concerns with regards to the results of the main outcomes. There were concerns

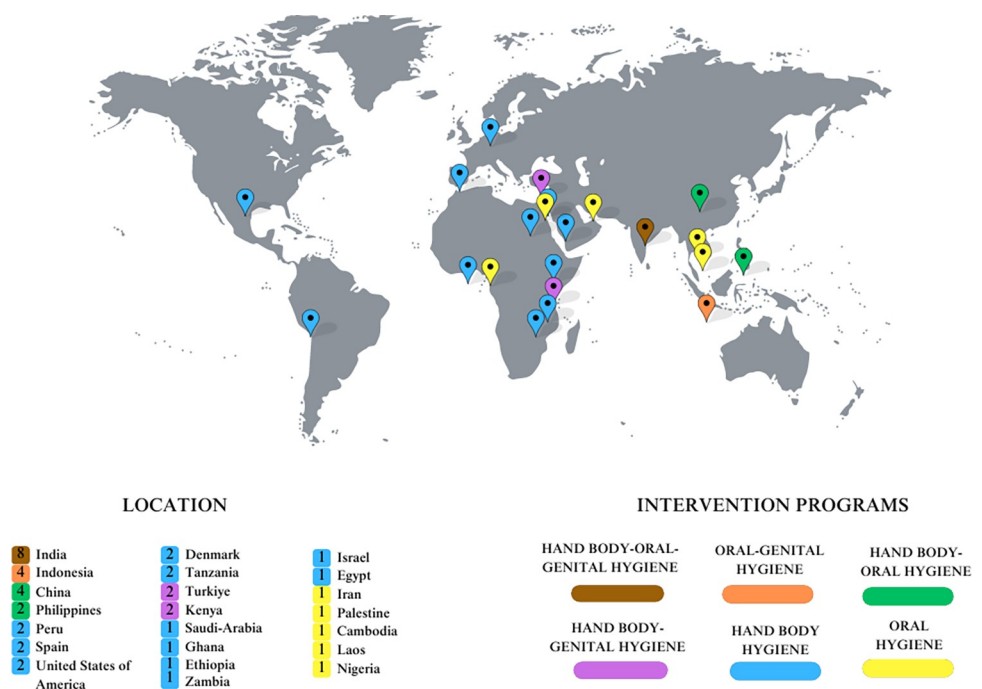

**Fig 3. Geographical representation of included studies.** Each location is represented by the types of interventions and total number of studies conducted.

in most studies in the following domains that they might lead to bias in the results: randomisation process, deviation from the intended interventions and measurement of the outcome.

Among 6 individual RCTs, 4 were judged to have an overall high risk-of-bias and 2 as having low risk-of-bias. The main domains of concern included deviation from the intended interventions and measurement of the outcome. The single cross-over trial was judged to have some concerns overall, with bias arising from the cross-over effects and deviation from the intended interventions being the domains of concern (S3 Appendix).

## Effect estimates

**Knowledge, attitudes and practices.** Students who participated in the hygiene intervention programs may have improved scores in overall knowledge, attitudes and practices (KAP) as compared to the control groups that underwent standard curriculum, although the evidence is presented with great uncertainty (SMD 2.30, 95%CI 1.17 to 3.44, 6 studies, 7301 participants, very-low-certainty-evidence) (Fig 4). All other forest plots of comparisons are found in S2 Appendix while the GRADE summary of findings is in Table 1.

Given the considerable heterogeneity among studies and conflicting direction of effects observed in Fig 4, we reassessed the intervention effects to the overall KAP score, specifically focusing on interventions with durations exceeding one month. The subgroup analysis revealed smaller improvements in overall KAP scores in the intervention group participants than in the control group (SMD 0.87, 95%CI 0.31 to 1.42, 5 studies; Analysis 1.2, S2 Appendix). Appiah-Brempong 2020 has the shortest overall knowledge, attitudes and practices intervention duration of two weeks [46] as compared to other studies with longer duration of 11 weeks [28, 29] to 2 years [44].

**Handwashing.** Handwashing practices during two crucial timings were documented in the included studies: after toilet and before meals. Students in the intervention groups may be more likely to perform handwashes (with or without soap) after toilet and before meals than students in the control group (RR1.75, 95%CI 1.41 to 2.17, 5 studies, 5479 participants; low-certainty- evidence (Fig 5).

**School absenteeism due to infections.** Students who underwent hand-body hygiene interventions may have a lower risk of absenteeism attributable to infections as compared to the standard curriculum (RR 0.74, 95%CI 0.66 to 0.83, 5 studies, 1017852 observations; low-certainty-evidence (Fig 6).

## Genital hygiene interventions compared to standard curriculum

Four studies reported genital hygiene interventions: Kenya (n = 1 study) [47], India (n = 1 study) [48], Indonesia (n = 1) [49] and Turkiye (n = 1 study) [33]. Three of the studies focused

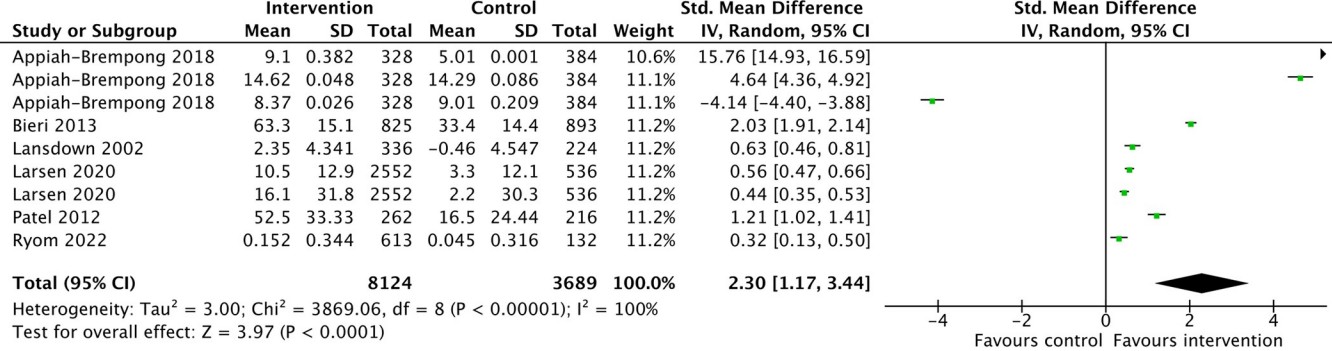

**Fig 4. Overall knowledge, attitudes, and practices of hand hygiene interventions.**

**Table 1. GRADE summary of hand hygiene interventions compared to standard curriculum in hand hygiene for school students.**

| Outcomes | № of participants (studies) Follow-up | Certainty of the evidence (GRADE) | Relative effect (95% CI) | Anticipated absolute effects | |
|---|---|---|---|---|---|
| | | | | Risk with SC | Risk difference with hand hygiene interventions |
| **Knowledge, attitudes and practices of hand hygiene** | 7301 (6 RCTs) | ⊕◯◯◯ Very low[1,2,3] | - | - | SMD **2.3 SD higher** (1.17 higher to 3.44 higher) |
| Assessed with knowledge, attitudes and practices surveys | | | | | |
| Follow-up: range 2 weeks to 2 years | | | | | |
| **Handwashing practices** | 5479 (5 RCTs) | ⊕⊕◯◯ Low[2] | **RR 1.75** (1.41 to 2.17) | 673 per 1,000 | **505 more per 1,000** (276 more to 788 more) |
| Assessed with number of handwashing events before eating and after toilet | | | | | |
| Follow-up: range 2 weeks to 1 years | | | | | |
| **School absenteeism due to infections** | 1017852* (5 RCTs) | ⊕⊕◯◯ Low[2] | **RR 0.74** (0.66 to 0.83) | 20 per 1,000 | **5 fewer per 1,000** (7 fewer to 3 fewer) |
| Assessed with self-reported or medically documented school absenteeism due to any infection | | | | | |
| Follow-up: range 5 weeks to 8 months | | | | | |

The risk in the intervention group (and its 95% confidence interval) is based on the assumed risk in the comparison group and the relative effect of the intervention (and its 95% CI).

RR–risk ratio; SC–standard curriculum; SMD–standardised mean difference.

*Presented as total number of observations instead of number of participants.

[1] Downgraded one level due to high risk of bias (Measurement outcome domain)

[2] Downgraded two levels due to very serious concerns in inconsistency.

on menstrual hygiene in girls. The characteristics of the included studies are summarised in S2 Table.

With great uncertainty, there may be an improvement in the overall attitudes toward genital hygiene in the intervention groups as compared to the standard curriculum (SMD 6.53, 95%CI 2.40 to 10.66, 2 studies, 2644 participants, very-low-certainty-evidence) (Table 2) (Analysis 2.1 in S2 Appendix).

There may also be an improvement in genital hygiene practices in students in the interventions group as compared to the students who received standard curriculum (RR 2.44, 95%CI 1.28 to 4.68, 1 study, 1201 participants, low-certainty-evidence) (Table 2) (Analysis 2.2 in S2 Appendix).

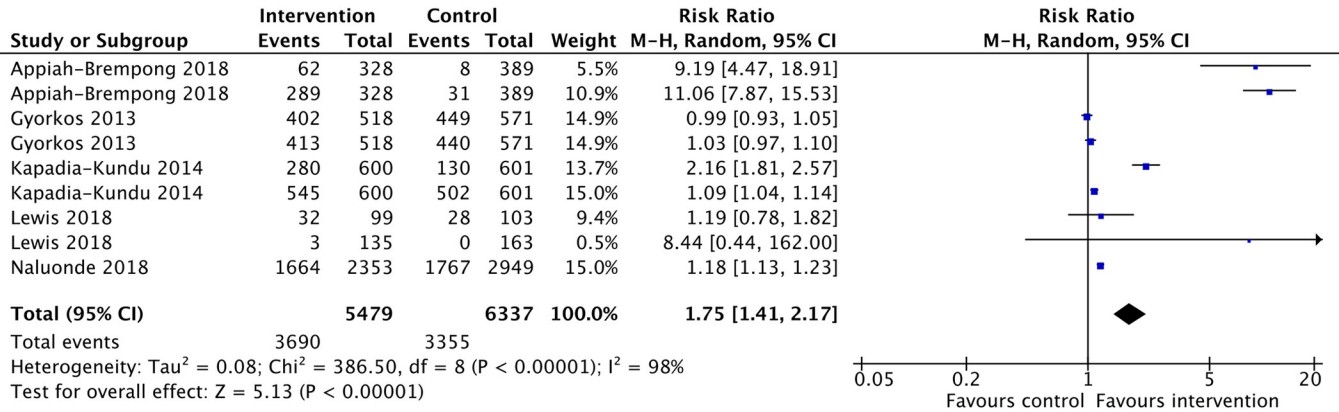

**Fig 5. Handwashing practices.**

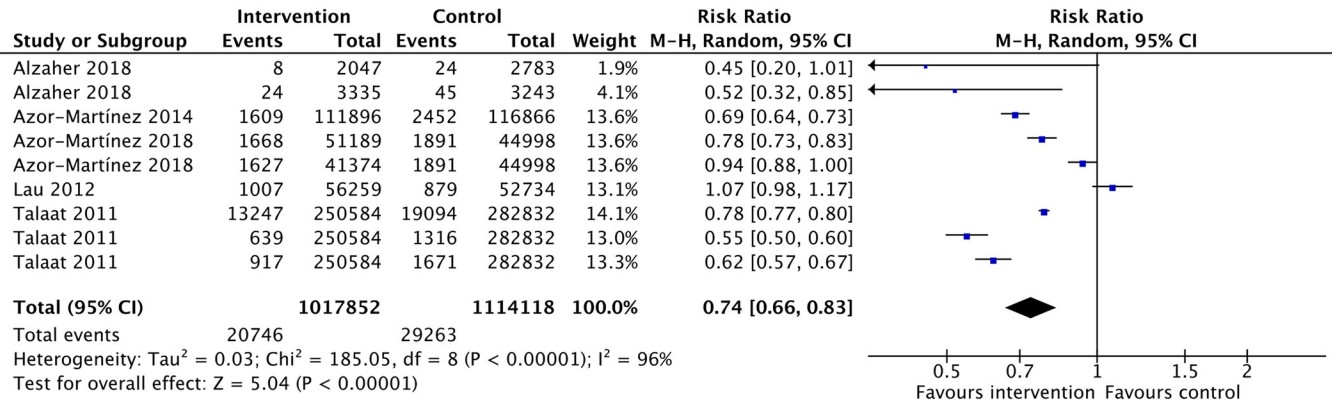

**Fig 6. School absenteeism due to infections.**

## Oral hygiene interventions compared to standard curriculum

There were ten trials (11 reports) reporting oral hygiene interventions compared to standard curriculum. Countries in which the trials were conducted include Cambodia (n = 1 study) [40], Indonesia (n = 3 studies) [40, 50, 51], India (n = 3) [37, 52, 53], Palestine (n = 1) [54], Lao PDR (n = 1 study) [40], Nigeria (n = 1 study) [51], Philippines (n = 1 study) [55], Iran (n = 1 study) [56], and China (n = 2) [30, 57].

Overall, four standardised scoring systems were used to assess oral health conditions: 1) Oral Hygiene Index Simplified (OHIS) for overall oral hygiene status; 2) Modified Gingival Index (MGI) for gum health status; 3) International Caries Detection and Assessment System (ICDAS) for dental caries; and 4) plaque score. ICDAS II 1–6 MFS and ICDAS II MFT were reported for the number of surfaces and number of teeth with dental caries (both nonactivated and cavitated lesions) [30, 57]. ICDAS II 3–6 MFS and ICDAS II 3–6 MFT were reported for

**Table 2. GRADE summary of genital hygiene interventions compared to standard curriculum in genital hygiene for school students.**

| Outcomes | № of participants (studies) Follow-up | Certainty of the evidence (GRADE) | Relative effect (95% CI) | Anticipated absolute effects | |
|---|---|---|---|---|---|
| | | | | Risk with SC | Risk difference with genital hygiene interventions |
| **Attitude changes in genital hygiene** <br> Assessed with genital hygiene attitude surveys <br> Follow-up: range 1 month to 17 months | 2644 (2 RCTs) | ⊕◯◯◯ <br> Very low[1,2,3] | - | - | SMD **6.53 SD higher** (2.4 higher to 10.66 higher) |
| **Genital hygiene practices** <br> Assessed with genital hygiene practices surveys <br> Follow-up: 1 year | 1201 (1 RCT) | ⊕⊕◯◯ <br> Low[1,3] | **RR 2.44** (1.28 to 4.68) | 191 per 1,000 | **276 more per 1,000** (54 more to 704 more) |

The risk in the intervention group (and its 95% confidence interval) is based on the assumed risk in the comparison group and the relative effect of the intervention (and its 95% CI).

RR–risk ratio; SC–standard curriculum; SMD–standardised mean difference.

[1] Downgraded one level due to high risk of bias (Measurement outcome domain)

[2] Downgraded two levels due to very serious concerns in inconsistency.

[3] Downgraded one level due to serious concerns in imprecision.

**Table 3. GRADE summary of oral hygiene interventions compared to standard curriculum in oral hygiene for school students.**

| Outcomes | № of participants (studies) Follow-up | Certainty of the evidence (GRADE) | Relative effect (95% CI) | Anticipated absolute effects | |
|---|---|---|---|---|---|
| | | | | Risk with SC | Risk difference with oral hygiene interventions |
| **Oral hygiene status** <br> Assessed with OHIS score, Plaque score, MGI score <br> Follow-up: range 6 months to 24 months | 652 (2 RCTs) | ⊕○○○ <br> Very low[1,2] | - | - | SMD **3.12 SD higher** (1.87 higher to 4.37 higher) |
| **Dental hygiene status** <br> Assessed DFMS score and DFMT score <br> Follow-up: range 1 year to 2 years | 4824 (3 RCTs) | ⊕⊕○○ <br> Low[1] | - | - | SMD **0.33 SD lower** (0.53 lower to 0.13 lower) |
| **Dental caries status** <br> Assessed with PUFA, ICDAS II 1–6 MFT, ICDAS II 3–6 MFT, ICDAS II 1–6 MFS, ICDAS II 3–6 MFS <br> Follow-up: range 1 year to 2 years | 2352 (4 RCTs) | ⊕⊕○○ <br> Low[1] | - | - | SMD **0.34 SD lower** (0.52 lower to 0.16 lower) |

The risk in the intervention group (and its 95% confidence interval) is based on the assumed risk in the comparison group and the relative effect of the intervention (and its 95% CI).

RR–risk ratio; SC–standard curriculum; SMD–standardised mean difference.

*Presented as total number of observations instead of number of participants.

[1] Downgraded two levels due to very serious concerns in inconsistency.

[2] Downgraded one level due to serious concerns in imprecision.

the number of surfaces and number of teeth with obvious dental caries (at cavitated level) [30, 57].

**Oral hygiene status.** Oral hygiene status was assessed in four reports from three trials [30, 36, 37, 57] using at least one of the following instruments: OHIS score, Plaque score and MGI score (Table 3). Findings of one study [37], could not be quantitatively synthesised with other studies due to lack of reporting.

With interventions lasting between 3 months and 6 months, there may be an overall worsening of oral hygiene, as indicated in an increase in oral hygiene scores in the intervention groups (SMD 3.12, 95%CI 1.87 to 4.37, 652 participants, low-certainty- evidence) (Fig 7) (Table 3).

Among the three studies included, one exhibited effect in the opposite direction. To explore the factors influencing this discrepancy, we examined distinctive characteristics of the included studies and found that the age group of the study population may have been a contributing factor to these divergent effects. In analysis 3.2 of S2 Appendix, only interventions conducted in secondary school students revealed improvements in dental hygiene indexes (SMD -0.27, 95%CI -0.39 to -0.15, 2 studies) (p = 0.001 for subgroup differences).

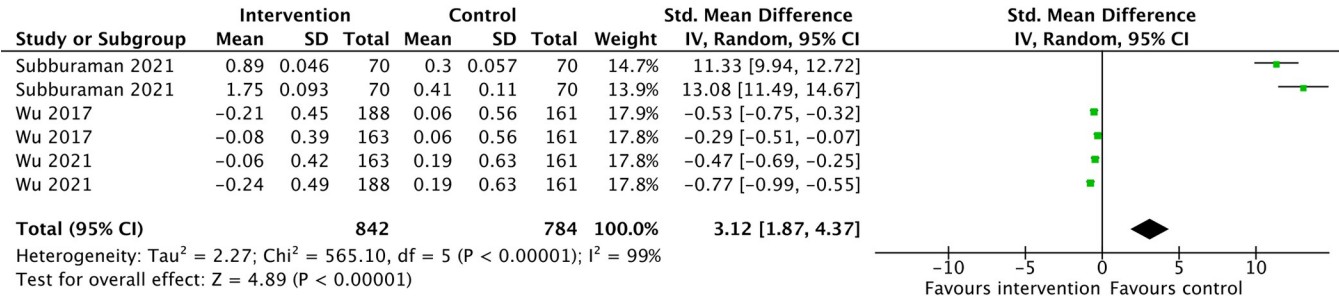

**Fig 7. Oral hygiene status.**

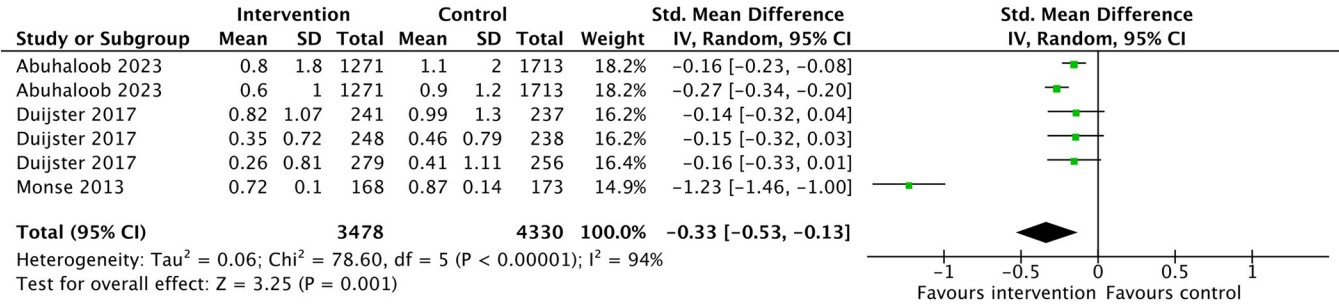

**Fig 8. Dental hygiene status.**

**Dental hygiene status.** Three studies from two trials reported comparisons between oral hygiene interventions and standard curriculum. These were conducted in Palestine [54], a combined study consisting of 3 countries i.e. Cambodia, Indonesia and Lao PDR (34) and another study done in the Philippines (52).

In comparing, oral health interventions with standard curriculum, changes in dental hygiene were reported using: 1) Decayed, Missing, and Filled Teeth/Surfaces (DFMT/DFMS) index for dental caries. There may be a modest reduction in dental caries in the intervention group (SMD -0.33, 95%CI -0.53 to -0.13, 3 studies, 4824 participants, low-certainty-evidence) (Table 3)(Fig 8).

**Dental caries status.** Three studies assessed dental caries score changes using: 1) Pulpal involvement, Ulceration, Fistula, and Abscess (PUFA) index for clinical complications of dental caries, 2) ICDAS II 1–6 MFT, 3) ICDAS II 3–6 MFT, 4) ICDAS II 1–6 MFS, and 5) ICDAS II 3–6 MFS.

These interventions were conducted in China [30, 57] while another was conducted in Cambodia, Indonesia and Lao PDR [40].

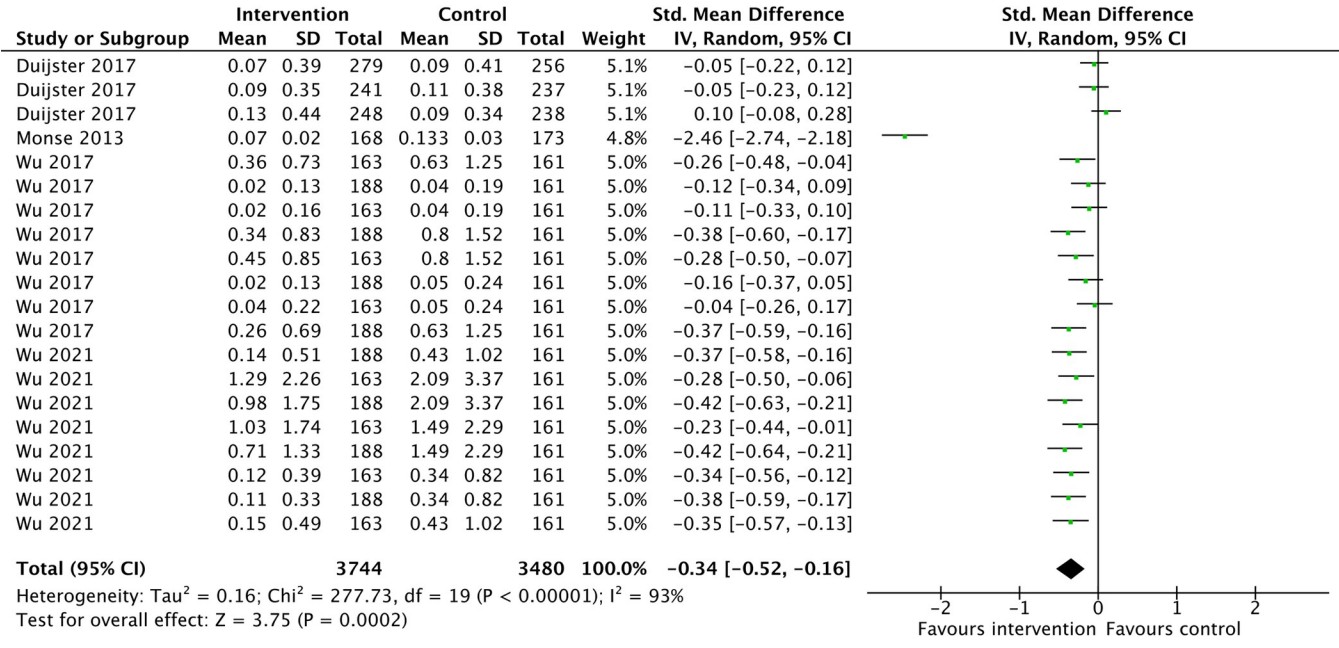

**Fig 9. Dental caries status.**

Reduction in dental caries was observed to be greater in the intervention groups than in the standard curriculum groups: (MD -0.34, 95%CI -0.52 to -0.16, 4 studies, 2352 participants, low-certainty-evidence) (Fig 9).

### Awareness of antimicrobial resistance

There is one study that evaluated antimicrobial resistance, which did not provide suitable outcome data for meta-analysis. The study was a non-randomised intervention study in high school students which compared a 15-min lecture on antimicrobial resistance and antibiotic used using United States Centres for Disease and Prevention and Control and the World Health Organisation materials, with standard curriculum [32].

The intervention group demonstrated significant improvements of participants' knowledge of antibiotics and causes of antibiotics resistance before and after intervention as compared to the control group (p<0.001). There were also significant improvements of participants' perceptions towards antibiotic resistance in the intervention group as compared to the control group (p<0.001).

## Discussion

### Summary of the main results

In this review, we found very low-to-low-certainty evidence that school-based interventions may improve overall knowledge, attitude and practice on general, hand and genital hygiene and reduce school absenteeism. The evidence on oral and dental hygiene status was mixed when measured using different tools. However, in most estimates, the effects varied among studies as indicated by high heterogeneity, and this was inadequately explained. Besides heterogeneity, the evidence is affected by the risk-of-bias of the studies and imprecision, as some outcomes were contributed by the small number of studies. These factors have led to a reduction in the certainty of the evidence presented.

### Agreement and disagreement with similar reviews

At least eight published reviews have evaluated school-based hygiene-related interventions. However, as far as we are aware, none of the published reviews covered the interventions in the same scope as the current review. Two reviews, each of 8 studies, that evaluated hand hygiene promotion interventions in schools reported potential modest benefits, but with low-quality evidence due mainly to study methodological limitations and conflicting results among included studies [58, 59]. Two reviews, including a Cochrane review that evaluated the use of rinse-free sanitizers in schools reported reduced school absenteeism in the intervention group but low-quality of evidence due to methodological shortcomings of the studies included [60, 61]. A review of 3 studies found suboptimal adherence to hand hygiene in schools (<50%), with educators demonstrating higher adherence than students [62]. School-based oral health interventions in India (10 articles) consistently reported improvements in oral hygiene among school children following the interventions. However, the evidence was limited due to methodological concerns in the included studies [63]. Evidence on Water, Sanitation and Hygiene (WASH) initiatives in schools was mixed, with some studies showing improvements in knowledge, attitude and behaviours, but no consistent significant differences in health outcomes like diarrhoea [23].

A broad Cochrane review on the WHO Health Promoting School framework covering healthy lifestyles found insufficient evidence on the effects of intervention on hand washing practice and oral health, with studies having low-to-moderate quality evidence and high risk of

bias [64]. Compared to the aforementioned reviews that evaluated a specific intervention, for example, hand hygiene or oral health promotion, the current review is broader in scope and encompasses all hygiene-related school-based interventions. Compared with other reviews that evaluated comprehensive healthy lifestyle intervention, our review is narrower in scope, focussing only on hygiene related interventions. However, the findings of our review are broadly in agreement with the aforementioned reviews, in showing modest benefit of low-to-very-low certainty for some of the hygiene related interventions and mixed findings for other interventions, highlighting methodological limitations of the included studies.

## Strengths and limitations of the review

We performed comprehensive searches from multiple databases and gathered a globally representative sample of studies conducted from 2011 to 2023 across 22 countries in America, Europe, Africa and Asia, making the evidence widely applicable. The systematic review was conducted using established methodologies, including the most updated version of the risk-of-bias tool and certainty-of-evidence assessments.

We acknowledge the following limitations. The main issue of concern in the review is the marked heterogeneity in the pooled estimates. Possible sources for the high degree of heterogeneity may be the specific details in population characteristics, intervention design, in particular, specific components of the intervention and comparator, including settings, methods, duration of administration and specific timing of assessments. More in-depth assessments of these factors are required, including among others, adherence to interventions and the reliability of self-reported outcome measures. However, these aspects were often not clearly defined and documented in the included studies. Next, the current review might not have captured a complete set of relevant studies. Despite a careful assessment of a broad set of nearly 500 studies that appeared relevant, we might have missed eligible studies, for instance, studies that evaluated interventions predominantly implemented outside the school setting or not clearly labelled as school-based but contained important components in the school. Next, in dealing with multiple reports, we performed careful assessments and merged the reports based on the reported study period, participant characteristics and settings. Despite our careful assessment, there remains a possibility of inappropriate combination of studies. Next, we deviated from our protocol in the following aspects: first, in the protocol, we included schoolteachers as our participants. In the review, we decided to focus on the students and exclude schoolteachers, as we did not find any paper that evaluated teachers' perception or knowledge following our current search strategies. To include studies that evaluated only teachers would probably have required a revision of our search strategies. Next, we decided to focus on the studies that evaluated the effects of hygiene intervention, rather than including non-interventional studies that evaluated knowledge, attitude, and practice, as well as the prevalence of hygiene-related practice. Although the protocol deviation might have led mainly to a narrowing of the scope of our evidence, there is a possibility that by excluding pre-specified participants and study types, we might have excluded relevant evidence.

Additionally, we acknowledge that the considerable variability in study design, implementation and evaluation of the school-based hygiene interventions impact the applicability of the study findings. Firstly, the interventions range from comprehensive multidimensional interactive sessions to structural changes to the school such as installing handwashing stations. The variability reflects the different contexts and needs of the study populations but would also limit its interpretation and comparison of findings across studies. Secondly, the duration and intensity of the interventions varied widely across the studies. Duration is an important predictor of behavioural interventions. Longer interventions generally provide more time for the

students to learn, practice, and reinforce new behaviours, which is a crucial for habit formation and skill development. To reduce these effects, we performed several sub-group analyses based on duration of interventions. This allowed for a more comparable outcome across different studies. Thirdly, the methods used to evaluate knowledge, attitude and practices (KAP) differed across the studies. Although this raises concerns about the ability to draw definitive conclusions from varying tools, it is important to note that KAP questionnaires should be tailored to the study population. This ensures relevance to the age, maturity, cultural, social, and environmental contexts of the study populations. Tailored questionnaires provide more meaningful and accurate data, which enhances the validity and reliability of the findings.

While we do recognise the importance of clear and thorough documentation and reporting practices in research, we believe that the large majority of the included studies demonstrated detailed methodologies, particularly in the assessment of outcomes. Therefore, poor documentation and reporting practices as a critical issue in this study. Nonetheless, we acknowledge that any instances of poor reporting and documentation of the included studies could impact the reliability of the outcomes and thus necessitate interpretation with caution.

While this study focuses on school-based interventions, we acknowledge the valuable insights provided by community-based interventions in child and adolescent health. Community-based programs also play a significant role in promoting healthy behaviours and can complement school-based efforts. However, the structured environment of schools offers unique opportunities for consistent and supportive interactions that are essential for habit formation and long-term behaviour change. Therefore, our emphasis on school-based interventions highlights their critical role in the holistic development of children.

## Addressing methodological limitations in future research

We had previously acknowledge that methodological limitations of the included studies contribute to the overall uncertainty of our findings. The variability in intervention design and implementation, including differences in duration and delivery methods, poses challenges for drawing definitive conclusions and comparing outcomes across studies. Additionally, many studies had a high or unclear risk of bias in key domains, such as the randomisation process and measurement of outcomes, which can affect the reliability of the results. The varying measurement tools across the studies also contributed to the heterogeneity of the findings. This limits the comprehensive assessment of the interventions.

Based on the limitations previously described, future studies should be aimed at providing a more robust evidence for the effectiveness of school-based hygiene interventions. This includes standardisation of intervention protocols, standardisation of measurement tools for different outcomes and specific for different age groups, and longer follow-up periods to assess the sustained impact of interventions. Addressing these issues will enhance the quality and comparability of the findings.

## Conclusion

Very-low-to-low-certainty-evidence shows that among school children, educational interventions targeting hand hygiene practice may improve knowledge and reduce school absenteeism due to respiratory and diarrheal illness. Genital hygiene educational intervention may improve knowledge and attitude based on mostly low-certainty-evidence. However, the effects of intervention on oral hygiene-related outcomes appeared mixed. Future research should strengthen randomisation and intervention documentation, and evaluate the association between knowledge gain, hygiene-related behaviour and health outcomes including school absenteeism, school academic performances and severity of infections.

## Supporting information

**S1 Table. Characteristics of studies with hand-body hygiene intervention programs.**
(DOCX)

**S2 Table. Characteristics of studies with genital hygiene intervention programs.**
(DOCX)

**S3 Table. Characteristics of studies with oral hygiene intervention programs.**
(DOCX)

**S1 Appendix. Search strategy.**
(DOCX)

**S2 Appendix. Summary of all meta-analysis.**
(DOCX)

**S3 Appendix. Summary of risk of bias assessment of included studies.**
(DOCX)

## Acknowledgments

We would like to thank the Director General of Ministry of Health Malaysia for his permission to publish this manuscript.

## Author Contributions

**Conceptualization:** Priya Madhavan, Esther van der Werf, Nai Ming Lai.

**Data curation:** Sophia Rasheeqa Ismail, Ranina Radzi, Puteri Sofia Nadira Megat Kamaruddin, Ezarul Faradianna Lokman, Han Yin Lim, Daarshini Arumugam, Alex Ngu, Annie Ching Yi Low, Eng Hwa Wong, Sapna Patil, Nai Ming Lai.

**Formal analysis:** Sophia Rasheeqa Ismail, Ranina Radzi.

**Investigation:** Sophia Rasheeqa Ismail, Annie Ching Yi Low, Eng Hwa Wong, Sapna Patil.

**Methodology:** Nusaibah Abdul Rahim, Hui Yin Yow, Daarshini Arumugam, Alex Ngu, Eng Hwa Wong, Sapna Patil, Priya Madhavan, Ruslin Bin Nordin, Esther van der Werf, Nai Ming Lai.

**Project administration:** Sophia Rasheeqa Ismail.

**Supervision:** Sophia Rasheeqa Ismail.

**Writing – original draft:** Sophia Rasheeqa Ismail, Ranina Radzi, Puteri Sofia Nadira Megat Kamaruddin, Ezarul Faradianna Lokman, Han Yin Lim, Nai Ming Lai.

**Writing – review & editing:** Sophia Rasheeqa Ismail, Ranina Radzi, Puteri Sofia Nadira Megat Kamaruddin, Ezarul Faradianna Lokman, Han Yin Lim, Nusaibah Abdul Rahim, Hui Yin Yow, Daarshini Arumugam, Alex Ngu, Annie Ching Yi Low, Eng Hwa Wong, Sapna Patil, Priya Madhavan, Ruslin Bin Nordin, Esther van der Werf, Nai Ming Lai.

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
