## [Decision Letter · Decision Letter 0]

6 May 2024

PONE-D-24-15555The effects of school-based hygiene intervention programme: Systematic Review and Meta-AnalysisPLOS ONE

Dear Dr. Ismail

Thank you for submitting your manuscript to PLOS ONE. After careful consideration, we feel that it has merit but does not fully meet PLOS ONE’s publication criteria as it currently stands. Therefore, we invite you to submit a revised version of the manuscript that addresses the points raised during the review process. Please submit your revised manuscript by Jun 20 2024 11:59PM. If you will need more time than this to complete your revisions, please reply to this message or contact the journal office at plosone@plos.org. Please include the following items when submitting your revised manuscript:A rebuttal letter that responds to each point raised by the academic editor and reviewer(s). You should upload this letter as a separate file labeled 'Response to Reviewers'.A marked-up copy of your manuscript that highlights changes made to the original version. You should upload this as a separate file labeled 'Revised Manuscript with Track Changes'.An unmarked version of your revised paper without tracked changes. You should upload this as a separate file labeled 'Manuscript'.If applicable, we recommend that you deposit your laboratory protocols in protocols.io to enhance the reproducibility of your results. Protocols.io assigns your protocol its own identifier (DOI) so that it can be cited independently in the future. For instructions see: https://journals.plos.org/plosone/s/submission-guidelines#loc-laboratory-protocols. Additionally, PLOS ONE offers an option for publishing peer-reviewed Lab Protocol articles, which describe protocols hosted on protocols.io. Read more information on sharing protocols at https://plos.org/protocols?utm_medium=editorial-email&utm_source=authorletters&utm_campaign=protocols.

We look forward to receiving your revised manuscript.

Kind regards,

Muhammad Shahzad Aslam, Ph.D.,M.Phil., Pharm-D

Academic Editor

PLOS ONE

Journal Requirements:

2. We note that Figure 3 in your submission contain map images which may be copyrighted. All PLOS content is published under the Creative Commons Attribution License (CC BY 4.0), which means that the manuscript, images, and Supporting Information files will be freely available online, and any third party is permitted to access, download, copy, distribute, and use these materials in any way, even commercially, with proper attribution. For these reasons, we cannot publish previously copyrighted maps or satellite images created using proprietary data, such as Google software (Google Maps, Street View, and Earth). For more information, see our copyright guidelines: http://journals.plos.org/plosone/s/licenses-and-copyright.

1) You may seek permission from the original copyright holder of Figure 3 to publish the content specifically under the CC BY 4.0 license.  

2) If you are unable to obtain permission from the original copyright holder to publish these figures under the CC BY 4.0 license or if the copyright holder’s requirements are incompatible with the CC BY 4.0 license, please either i) remove the figure or ii) supply a replacement figure that complies with the CC BY 4.0 license. Please check copyright information on all replacement figures and update the figure caption with source information. If applicable, please specify in the figure caption text when a figure is similar but not identical to the original image and is therefore for illustrative purposes only.

3. We note that this manuscript is a systematic review or meta-analysis; our author guidelines therefore require that you use PRISMA guidance to help improve reporting quality of this type of study. Please rename the completed PRISMA checklist as Supporting Information with a file name “PRISMA checklist”.

**Additional Editor Comments:**

1. Various studies show diverse approaches in how interventions are designed, implemented, and evaluated, impacting the applicability of findings. It's crucial to address this variability in the main text.

2. Poor documentation and reporting practices obscure a clear understanding of research outcomes, underscoring the importance of addressing this issue within the main text.

3. Focusing solely on interventions within school settings may overlook valuable insights from community-based studies. Integrating a brief acknowledgment of this in the main text could enhance the comprehensiveness of the analysis.

4. Any deviations from standard practices, like excluding key stakeholders such as schoolteachers or narrowing the scope, can potentially compromise the integrity and completeness of the study. This warrants further explanation within the main text.

5. Methodological limitations contribute to overall uncertainty, highlighting the necessity for additional research endeavors. Elaborating on these limitations within the main text can underscore the imperative for future investigations.

Reviewers' comments:

Reviewer's Responses to Questions

**Comments to the Author**

1. Is the manuscript technically sound, and do the data support the conclusions?

Reviewer #1: Yes

Reviewer #2: Yes

Reviewer #3: Yes

2. Has the statistical analysis been performed appropriately and rigorously? 

Reviewer #1: Yes

Reviewer #2: Yes

Reviewer #3: Yes

3. Have the authors made all data underlying the findings in their manuscript fully available?

Reviewer #1: Yes

Reviewer #2: Yes

Reviewer #3: Yes

4. Is the manuscript presented in an intelligible fashion and written in standard English?

Reviewer #1: Yes

Reviewer #2: Yes

Reviewer #3: Yes

5. Review Comments to the Author

Reviewer #1: The authors have done a comprehensive study and have provided a clear discussion about their findings. However, some clarifications are needed to clarify about their study. Some of the clarifications needed are:

1) Suggest for the authors to clearly mention the types of college that they have taken into the inclusion criteria (Line 120)

2) The justification in the exclusion criteria for not including healthcare-related colleges is rather weak. Can the authors elaborate why they have excluded this criteria?

3) Why authors considered self-reported causes documentation as part of their study? This type of documentation is exposed with self-claimed of one's health status to avoid from attending classes.

4) For Line 137, antibiotics prescription is only applicable for bacterial infection. Have the authors considered about viral infections as well?

5) For Line 150, how did authors deal with any manuscripts that were not written in English or the authors' mother-tongue?

6) For Line 177, on what basis that a manuscript was being judged as 'Low', 'Some concerns' etc? Any criteria during this judgement process?

Reviewer #2: I would like to express my gratitude to the authors for your comprehensive and insightful systematic assessment of the effects of school-based hygiene intervention programmes. Your study sheds light on the effectiveness of such interventions, with major implications for practice and future research.

Reviewer #3: The research titled “The effects of school-based hygiene intervention programme: Systematic Review and Meta-Analysis”

General comments:

1. The review encompasses 39 trials from 22 countries, offering a comprehensive assessment of global school-based hygiene interventions.

2. Meta-analysis suggests hand-body and genital hygiene interventions improve knowledge, attitudes, and practices, reducing infection-related absenteeism among school children.

3. Rigorous methodologies, including risk-of-bias assessment and certainty-of-evidence ratings, ensure reliable and applicable findings.

Addressing the following comments:

1. High variation in intervention design, duration, and assessment methods across studies limits generalizability. In this regard, there is a need to explain in the main text.

2. Inadequate documentation and reporting hinder comprehensive understanding of results. In this context, there is a need to explain in the main text.

3. Focusing solely on school-based interventions may overlook relevant community studies. In this context, there is a need to add 1-2 sentences in the main text.

4. Deviations, such as excluding schoolteachers and narrowing scope, may affect completeness. In this context, there is a need to explain in the text.

5. Overall, certainty remains low due to methodological limitations, emphasizing the need for further research. In this regard, there is a need to elaborate in the main text.

Report:

After addressing the suggested comments, this paper can be published in PLOS ONE.

6. PLOS authors have the option to publish the peer review history of their article (what does this mean?). If published, this will include your full peer review and any attached files.

Reviewer #1: **Yes: **MUHAMAD AFIQ FAISAL BIN YAHAYA

Reviewer #2: No

Reviewer #3: **Yes: **Vetriselvan Subramaniyan

---

## [Author Response · Author response to Decision Letter 0]

19 Jun 2024

Dear Reviewers,

Thank you for your thorough and insightful comments on our manuscript. We appreciate the time and effort you have invested in reviewing our work. We have carefully considered each of your suggestions and have made the following revisions in response to your feedback.

With this revision, we have submitted the following documents: 

1. Our revised manuscript with track changes labelled as ‘Revised Manuscript With Tracked Changes’

2. Our revised manuscript without track changes labelled as ‘Manuscript’

JOURNAL REQUIREMENTS:

Thank you for your comment. We have thoroughly reviewed PLOS ONE's style requirements and ensured that our manuscript now adheres to all specified guidelines. Specifically file naming, formatting (includes font type, size and line spacing), and reference format. 

2. We note that Figure 3 in your submission contain map images which may be copyrighted. All PLOS content is published under the Creative Commons Attribution License (CC BY 4.0), which means that the manuscript, images, and Supporting Information files will be freely available online, and any third party is permitted to access, download, copy, distribute, and use these materials in any way, even commercially, with proper attribution. For these reasons, we cannot publish previously copyrighted maps or satellite images created using proprietary data, such as Google software (Google Maps, Street View, and Earth). For more information, see our copyright guidelines: http://journals.plos.org/plosone/s/licenses-and-copyright.

Thank you for bringing this to our attention. We would like to clarify that the image included in Figure 3 is not copyrighted. It was created by our team based on our research findings using the free version of the software tool Canva (https://www.canva.com/), which is available for such purposes. The image was generated specifically to represent our findings accurately and does not use any proprietary data.

Furthermore, we confirm that this image can be freely used by future authors, promoting continued research and collaboration in the field. We have ensured that it complies with the Creative Commons Attribution License (CC BY 4.0). Please refer Figure 3. 

3. We note that this manuscript is a systematic review or meta-analysis; our author guidelines therefore require that you use PRISMA guidance to help improve reporting quality of this type of study. Please rename the completed PRISMA checklist as Supporting Information with a file name “PRISMA checklist”. 

Thank you for your comment. We have ensured that our manuscript adheres to the PRISMA (Preferred Reporting Items for Systematic Reviews and Meta-Analyses) guidelines to enhance the reporting quality of our systematic review and meta-analysis.

We have completed the PRISMA checklist and renamed the file as "PRISMA checklist" as per your instructions. This file is included as Supporting Information in our submission.

Thank you for your comment. We have thoroughly reviewed our reference list to ensure it is complete and accurate. 

ADDITIONAL EDITOR COMMENTS:

1. Various studies show diverse approaches in how interventions are designed, implemented, and evaluated, impacting the applicability of findings. It's crucial to address this variability in the main text.

Thank you for this remark. We have addressed the variability in the design, implementation, and evaluation of interventions in the Discussion section of the manuscript. We have also elaborated how differences in intervention design, implementation duration, and evaluation methods can influence the outcomes and their comparability across studies. (Please refer to lines 491 – 508)

2. Poor documentation and reporting practices obscure a clear understanding of research outcomes, underscoring the importance of addressing this issue within the main text.

Thank you for the query. While we do recognise the importance of clear and thorough documentation and reporting practices in research, we believe that the large majority of the included studies demonstrated detailed methodologies, particularly in the assessment of outcomes. Therefore, poor documentation and reporting practices as a critical issue in this study. Nonetheless, we acknowledge that any instances of poor reporting and documentation of the included studies could impact the reliability of the outcomes and thus necessitate interpretation with caution. (Please refer to lines 509 – 514)

3. Focusing solely on interventions within school settings may overlook valuable insights from community-based studies. Integrating a brief acknowledgment of this in the main text could enhance the comprehensiveness of the analysis.

Thank you for your comment. Our research focus on school-based interventions does not undermine the relevance of community-based interventions in child and adolescent health. We acknowledge the importance of community-based studies and their valuable insights. However, we believe that school-based interventions play a pivotal role in the holistic development of school-going children, which is why we have chosen to focus on this aspect in our manuscript.

Schools provide a unique environment for child and adolescent development, offering structured settings where they can have positive and supportive interactions. These interactions are crucial for fostering healthy development and establishing long-lasting habits which would lead to better hygiene practices and ultimately resulting in positive health impacts.

To address this concern, we have included a brief acknowledgment of the relevance of community-based interventions in the Discussion section of the manuscript. (Please refer to lines 515 – 521)

4. Any deviations from standard practices, like excluding key stakeholders such as schoolteachers or narrowing the scope, can potentially compromise the integrity and completeness of the study. This warrants further explanation within the main text.

We acknowledge the importance of including key stakeholders, such as schoolteachers, in studies to ensure comprehensive insights. However, in our systematic review, the included studies did not demonstrate any deviations from standard practices. Thus, we are in belief that this exclusion is not a concern for the integrity and completeness of our current analysis. However, future research should consider the role of schoolteachers and other stakeholders to provide a more comprehensive understanding of intervention impacts.

5. Methodological limitations contribute to overall uncertainty, highlighting the necessity for additional research endeavours. Elaborating on these limitations within the main text can underscore the imperative for future investigations.

We agree that methodological limitations contribute to overall uncertainty and that it is crucial to elaborate on these within the main text to underscore the need for future research. We have added a detailed discussion of methodological limitations, including issues related to study design, sample size, intervention implementation, and outcome measurement in the Discussion section. (Please refer to lines 522 – 536)

RESPONSES TO COMMENTS BY REVIEWER 1: 

1. Suggest for the authors to clearly mention the types of college that they have taken into the inclusion criteria (Line 120)

Thank you for your suggestion. We have revised the manuscript to clearly specify the types of colleges included and excluded in our study. (Please refer to lines 123 – 125). 

2. The justification in the exclusion criteria for not including healthcare-related colleges is rather weak. Can the authors elaborate why they have excluded this criteria?

Thank you for your comment. We appreciate the opportunity to clarify our exclusion criteria. Healthcare students, such as those studying nursing, medicine, or other health-related fields, are not representative of the general population or typical school-aged children for several reasons. 

Firstly, healthcare students receive extensive education and training in hygiene practices and the prevention of infections as part of their curriculum. Thus this prior knowledge significantly influences their hygiene behaviours and practices, making their baseline levels of knowledge and practices different from those of the general student population.

Secondly, healthcare students are also go through hygiene and infection control training throughout their curriculum and are required to have a higher standard of hygiene practices as compared to the general student population. 

And lastly, as the overall aim of this review is to evaluate the effectiveness of hygiene intervention programs among typical school-aged children, the inclusion of the healthcare students would skew the results of our study as their knowledge and practices are not representative of the general population. 

3. Why authors considered self-reported causes documentation as part of their study? This type of documentation is exposed with self-claimed of one's health status to avoid from attending classes.

We included self-reported causes of absenteeism in our study to ensure a comprehensive understanding of the impact of school-based hygiene interventions. Many illnesses, such as acute gastroenteritis, often resolve without formal medical treatment through home remedies or over-the-counter solutions. Excluding self-reported cases would result in significant underreporting of these instances, leading to a biased and incomplete picture of school absenteeism. Repeated reports from the same student also enhance the reliability of self-reported data. While we acknowledge that some self-reported absences might be due to reasons other than illness, such as avoiding classes, these instances are likely minimal. Schools typically have mechanisms to detect patterns of absenteeism due to non-medical reasons, ensuring that our data primarily reflect genuine health-related absenteeism. By considering both medically documented and self-reported causes, we aim to capture the full spectrum of absenteeism and better assess the effectiveness of hygiene interventions.

4. For Line 137, antibiotics prescription is only applicable for bacterial infection. Have the authors considered about viral infections as well?

Thank you for your comment. Yes, viral infections have been considered in our study, as reflected under the same point. We recognise that mild viral infections are often treated at home with home remedies or over-the-counter medications to alleviate symptoms. In cases where the viral infection is severe, it would typically be medically documented, as the students would seek medical treatment or be hospitalised. Our analysis accounts for both scenarios, ensuring a comprehensive understanding of absenteeism due to various types of pathogens, whether it is bacterial, viral, fungal or parasites. 

Additional information has been added to the sentence; “The types of infections included respiratory infections (characterised by symptoms such as sore throats, coughs, asthma, and influenza-like illness), gastrointestinal infections (characterised by symptoms such as diarrhoea and vomiting), and unspecified infections requiring prescription of antibiotics or other treatments.” (Please refer to lines 134 – 140)

5. For Line 150, how did authors deal with any manuscripts that were not written in English or the authors' mother-tongue?

We did not employ any language restrictions during selection of relevant articles in this review. Some articles had abstracts in dual languages. If we encounter any manuscripts not written in English or the author's mother-tongue, we will arrange for their translation. For example, we would have sought for translation assistance, ensuring that all relevant studies are accessible and included in our analysis. However, none of the potentially included articles (from full text screening) were written in a language other than English. Thus exclusion based on language was not performed.

6. For Line 177, on what basis that a manuscript was being judged as 'Low', 'Some concerns' etc? Any criteria during this judgement process?

The risk of bias assessment for our manuscript was conducted using The Revised Cochrane risk-of-bias tool for randomized trials (RoB 2) (Access to tools and guidelines can be found here: https://sites.google.com/site/riskofbiastool/welcome/rob-2-0-tool?authuser=0 ) .This tool evaluates risk of bias across five domains: the randomization process, deviations from intended interventions, missing outcome data, measurement of the outcome, and selection of the reported result. Each domain's risk was assessed based on specific criteria and signalling questions provided in the RoB 2 documentation. Studies were judged as 'Low Risk of Bias' if they met the criteria for low risk in all domains, 'Some Concerns' if there were concerns in at least one domain, and 'High Risk of Bias' if there was high risk in at least one domain or multiple domains with some concerns. Additionally, adjustments were made for specific interventions and circumstances to ensure a fair assessment. These criteria and the judgement process have been elaborated in the "Methods" section of the revised manuscript. (Please refer to lines 182 – 185). 

RESPONSES TO COMMENTS BY REVIEWER 3: 

1. High variation in intervention design, duration, and assessment methods across studies limits generalizability. In this regard, there is a need to explain in the main text.

2. Inadequate documentation and reporting hinder comprehensive understanding of results. In this context, there is a need to explain in the main text.

3. Focusing solely on school-based interventions may overlook relevant community studies. In this context, there is a need to add 1-2 sentences in the main text.

4. Deviations, such as excluding schoolteachers and narrowing scope, may affect completeness. In this context, there is a need to explain in the text.

5. Overall, certainty remains low due to methodological limitations, emphasizing the need for further research. In this regard, there is a need to elaborate in the main text.

All of these comments have been responded under the Additional Editors’ Comments section and elaborated with the necessary revisions made in the manuscript.

---

## [Decision Letter · Decision Letter 1]

16 Jul 2024

The effects of school-based hygiene intervention programme: Systematic Review and Meta-Analysis

PONE-D-24-15555R1

Dear Dr. Ismail,

We’re pleased to inform you that your manuscript has been judged scientifically suitable for publication and will be formally accepted for publication once it meets all outstanding technical requirements.

Kind regards,

Muhammad Shahzad Aslam, Ph.D.,M.Phil., Pharm-D

Academic Editor

PLOS ONE

Additional Editor Comments (optional):

Reviewers' comments:

Reviewer's Responses to Questions

**Comments to the Author**

1. If the authors have adequately addressed your comments raised in a previous round of review and you feel that this manuscript is now acceptable for publication, you may indicate that here to bypass the “Comments to the Author” section, enter your conflict of interest statement in the “Confidential to Editor” section, and submit your "Accept" recommendation.

Reviewer #1: All comments have been addressed

Reviewer #2: All comments have been addressed

2. Is the manuscript technically sound, and do the data support the conclusions?

Reviewer #1: Yes

Reviewer #2: Yes

3. Has the statistical analysis been performed appropriately and rigorously? 

Reviewer #1: Yes

Reviewer #2: Yes

4. Have the authors made all data underlying the findings in their manuscript fully available?

Reviewer #1: Yes

Reviewer #2: Yes

5. Is the manuscript presented in an intelligible fashion and written in standard English?

Reviewer #1: Yes

Reviewer #2: Yes

6. Review Comments to the Author

Reviewer #1: The authors have addressed all prior comments by providing comprehensive justifications. Also, the authors have amended the manuscript according to the given justifications. The manuscript is worth to be accepted and published.

Reviewer #2: (No Response)

7. PLOS authors have the option to publish the peer review history of their article (what does this mean?). If published, this will include your full peer review and any attached files.

Reviewer #1: **Yes: **MUHAMAD AFIQ FAISAL BIN YAHAYA

Reviewer #2: **Yes: **Dr.Sreenivasulu Sura

---

## [Editor Report · Acceptance letter]

24 Jul 2024

PONE-D-24-15555R1 

PLOS ONE

Dear Dr. Ismail, 

I'm pleased to inform you that your manuscript has been deemed suitable for publication in PLOS ONE. Congratulations! Your manuscript is now being handed over to our production team.

Kind regards, 

on behalf of

Dr. Muhammad Shahzad Aslam 

Academic Editor

PLOS ONE